# Comparison of Visual and Aberrometric Outcomes in Suture-Free Scleral Fixation: Three-Piece Lenses versus Carlevale Lenses

**DOI:** 10.3390/jcm12010188

**Published:** 2022-12-26

**Authors:** Fabrizio Franco, Federica Serino, Giulio Vicini, Cristina Nicolosi, Fabrizio Giansanti

**Affiliations:** 1Eye Clinic, Neuromuscular and Sense Organs Department, Careggi University Hospital, 50139 Florence, Italy; 2Department of Neurosciences, Psychology, Drug Research and Child Health, University of Florence, 50139 Florence, Italy; 3Azienda USL Toscana Nord Ovest, 56121 Pisa, Italy

**Keywords:** Carlevale lens, three-piece lens, scleral fixation, IOL tilt, refractive outcomes

## Abstract

Purpose: To compare the refractive results between sutureless scleral fixation intraocular lens (IOLs) (Carlevale, Soleko) and suture-free scleral fixation three-piece IOLs (Sensar AR40, Johnson & Johnson) for secondary implantation in patients with IOL dislocation or aphakia. Methods: This is a monocentric retrospective study on 28 patients (28 eyes) with sutureless scleral fixation Carlevale IOL and 25 patients (25 eyes) with suture-free scleral fixation three-piece IOL. Best-corrected visual acuity (BCVA) evaluation, refractive measures and IOL tilt evaluation with anterior segment optical coherence tomography were conducted at one, three, six and twelve months after surgery. Point Spread Function (PSF) was measured using a total ocular aberrometer. Results: BCVA in both groups improved since the postoperative visit at 1 month and reached a stable value at 3 months At month 12, mean BCVA was 0.23 logMAR in group one and 0.32 logMAR in group two. Mean IOL tilt angle at 12 months was 2.76° ± 1.87 in group one and 2.51° ± 1.80 in group two. PSF at 12 months was 0.18 ± 0.09 in group one and 0.15 ± 0.05 in group two. There were no statistically significant differences (*p* > 0.05) for all comparisons. The post-operative complications were similar within the two groups. Conclusions: Our results show that secondary IOL implantation has similar visual and surgical outcomes when a sutureless Carlevale lens scleral fixation and a suture-free scleral fixation three-piece IOL are used.

## 1. Introduction

Intraocular lenses (IOLs) implantation for aphakia correction could be a surgical challenge, especially in the absence of an effective capsular support. A leading cause for surgery is late dislocation of in-the-bag or out-of-the bag IOL (75%), followed by secondary aphakia (19%) and other causes (6%) [1]. The incidence of late IOL dislocation requiring secondary intervention varies among studies from 0.5% to 1.7% [2] Various techniques have been developed over the last years, for IOLs implant either in the anterior and in the posterior chamber. Traditionally, the secondary IOL implantation techniques employ rigid IOLs, which require wide corneal or scleral incisions with difficult intra- and/or post-operative management [3]. The wound construction is an important and well-known factor associated with postoperative astigmatism [4].

Recently, the increasing attention to the refractive results after cataract surgery has led to the research of procedures that try to avoid astigmatism, and to the introduction of foldable lenses which can be placed through smaller incisions. This allows a more rapid visual recovery, less post operative astigmatism, and better refractive outcomes. A Carlevale scleral fixated sutureless IOL (Soleko, Italy) is a one-piece foldable IOL specifically designed for posterior chamber implantation [5]: the main characteristic is the new “T-shaped” design of the plugs at the end of the two haptics which make it possible to anchor the IOL to the sclera without the need of sutures. Previous studies demonstrated long-term postoperative stability, and consequently good visual and refractive outcomes [6]. Recently, new techniques employing traditional posterior chamber three-piece IOLs, in which IOL haptics are externalized and fixated without sutures, have become popular due to the good visual outcome reported [7,8,9].

We conducted a retrospective study with the purpose of comparing visual and refractive outcomes between sutureless scleral fixation IOLs (Carlevale FIL-SSF, Soleko, Italy) and suture-free three-piece IOLs (Sensar AR40, Johnson & Johnson Vision, Irvine, CA, USA) for secondary IOL implantation. We aimed to analyze the clinical results and complication rate of these two suture-free IOLs, assuming that they are similar and may represent an alternative to each other.

## 2. Materials and Methods

This is a monocentric retrospective study on patients treated with scleral fixation IOL implantation between February 2020 and September 2021. The study was conducted at the Eye Clinic, Neuromuscular and Sense Organs Department, of Careggi University Hospital (Florence, Italy), and was performed according to the current version of the Declaration of Helsinki (52nd WMA General Assembly, Edinburgh, Scotland, UK, October 2000). All the patients included in the study signed a written informed consent, agreeing to participate. The study was approved by the Careggi University Hospital Research Ethics Board.

We included patients with aphakia after cataract extraction or lens luxation (traumatic or idiopathic) and patients with lens subluxation (traumatic or idiopathic). Exclusion criteria consisted of high refractive errors (myopia > 6 diopters, hypermetropia or astigmatism > 3 diopters), pupil diameter > 4.5 mm, significant ocular comorbidities, such as corneal lesions, retinal diseases, uveitis, and glaucoma.

At baseline each patient underwent a complete ophthalmic examination, including best-corrected visual acuity (BCVA) evaluation, intraocular pressure measurement with Goldmann applanation tonometry, ocular eye examination with biomicroscopy of the anterior segment and dilated fundus examination. The refractive assessment was obtained by means of autorefractometer (Autorefractor Keratometer, Nidek, Japan) and ocular aberrometer (Osiris, CSO Italy). Refraction measurements were expressed as spherical equivalent. The aberrometric evaluation was conducted by measuring the point spread function (PSF) in the central 3 mm. All the patients have been operated by the same experienced surgeon (F.F.) and with standardized procedures. All the surgeries were performed under locoregional anesthesia.

Patients were divided into two different treatment groups: the first group (Carlevale group) included patients undergoing the secondary implant of a single-piece sutureless scleral fixation IOL (Carlevale FIL-SSF IOLs, Soleko, Italy) and the second group (three-piece lens group) included patients undergoing the secondary implant of a three-piece IOL (Sensar AR40, Johnson & Johnson Vision, USA), anchored to the sclera without sutures. In both groups a 3 × 3 mm scleral flap was created to cover the extremities of the IOLs.

The Carlevale FIL-SSF IOL (Soleko, Italy) is a hydrophilic acrylic foldable lens with an overall diameter of 13.2 mm. It has been specifically created for sutureless scleral fixation: this can be possible thanks to the special and innovative design of the T-shaped trans-scleral plugs of the haptics, which can be anchored to the sclera without sutures. For the implant of the sutureless Carlevale IOL, the surgeon inserted the T-plugs through the sclerotomy and then externalized the plugs under the scleral flap with crocodile tip forceps.

The three-piece IOL Sensar AR40 (Johnson & Johnson, New Brunswick, NJ, USA) is an acrylic hydrophobic biconvex aspheric IOL; it has a 6 mm optic diameter and two Polymethyl Methacrylate (PMMA) haptics, with an overall length of 13 mm. The lens is foldable, and the thin and rigid configuration of the haptics makes this IOL suitable for scleral fixation, although it was initially designed for capsular and sulcus implantation. A scleral “flanged” fixation was performed. The sclera was marked in 2 points at 1.8 mm from the limbus, 180 degrees distant, then 2 sclerotomies were created by means of 25-gauge sclerotome in these points, under the scleral flap and the haptics were externalized through the sclerotomies using crocodile forceps (Figure 1). An accurate measurement of the 0–180 degrees distance between the sclerotomies is critical for achieving a correct IOL centration. Finally, the edge of the haptics were cauterized with low temperature cautery to make a bulb.

Postoperative clinical data, including postoperative complications occurrence, have been collected at the following time points after surgery: 1 month, 3 months, 6 months, and 12 months. PSF was measured at post-operative visits at month 3 and 12.

The tilting was evaluated on swept source anterior segment optical coherence tomography (AS-OCT) (MS39, CSO, Scandicci (Florence), Italy) scans at month 1, month 3, month 6 and month 12. The angle of tilt was measured as proposed by Yamane et al. [7]: pupil was dilated, then a radial scan was acquired by means of the swept source AS-OCT. Horizontal (0–180°) and vertical (90–270°) scans were analyzed: the reference line was considered a straight line passing through the iridocorneal angle, and a second line was drawn on the horizontal axis of the IOL (Figure 2). The tilt angle was defined as the angle between these two lines centered on the corneal apex. The average of the IOL tilt angle in the horizontal and vertical images was recorded. OCT macular scan was performed at every follow-up visit to detect post-operative macular edema.

Statistical analysis was performed using SPSS Statistics (SPSS Inc., Chicago, IL, USA) software for macOS (Version 26.0). Demographic and clinical characteristics of the two treatment groups were compared using a two-tailed Student’s *t*-test or Chi-square test with 95% confidence intervals. The chosen level of statistical significance was *p*-value < 0.05.

## 3. Results

Fifty-three patients (53 eyes) treated with secondary scleral-fixated IOL implantation were included in the study. Twenty-eight patients (28 eyes) were treated with the Carlevale lens implantation (group 1), and 25 patients (25 eyes) with the three-piece IOL Sensar AR40 implantation (group 2). The demographic and clinical characteristics of patients included in the study are summarized in Table 1.

Mean BCVA at baseline was 0.78 logMAR in grouOKp 1 and 0.81 logMAR in group 2. BCVA in both groups improved since the postoperative visit at month 1 and reached a stable value at month 3. At month 12, mean BCVA was 0.23 logMAR in group 1 and 0.32 logMAR in group 2. The mean postoperative refractive error at 12 months, expressed as spherical equivalent, was −0.33 D in group 1 and −0.35 D in group 2.

Mean IOL tilt angle at 12 months was 2.75° in group 1 and 2.51° in group 2. A significant IOL tilt (angle > 5°), occurred at a similar percentage within the two groups (7 eyes with a Carlevale lens and 6 eyes with a three-piece lens, 25% and 24% respectively). At post-operative visit at month 12, PSF was 0.18 in group 1 and 0.15 in group 2. There were no statistically significant differences (*p*-value > 0.05) for all comparisons. Postoperative results and comparison between the two groups are summarized in Table 2.

Regarding postoperative complications, no patients included in the study had lens displacement, meant as the edge of the IOL involving the visual axis and causing vision loss. Transient corneal edema was detected in 3 patients (12%) with the three-piece IOL and in 1 patient (3.6%) with the Carlevale lens, whereas bullous keratopathy requiring endothelial transplant occurred in one patient (3.6%) treated with the Carlevale lens. We found a single case with haptic exposure in group 1 (3.6%). Two patients in both groups (7.1% and 8%, respectively) had cystoid macular edema which disappears with topical non-steroidal anti-inflammatory (NSAID) and corticosteroid therapy. Postoperative complications are reported in Table 3.

## 4. Discussion

The capsular bag-IOL implantation represents the standard of care after cataract surgery, with an IOL position near to the physiological condition. Sometimes this approach is not possible, because of the lack of an effective capsular support which may occur because of preoperative conditions (e.g., zonular laxity, pseudoexfoliation syndrome), intraoperative complications (anterior or posterior capsular tears) or post operative events (spontaneous IOL luxation/subluxation, traumas). In these cases, the IOL must be placed in an alternative site.

Secondary IOL implantations became more frequent in recent years, with the increasing number of cataract surgeries performed and the progressive aging of the population. The surgical procedures are quickly evolving, and different options are currently available, ranging from implantation in the iridocorneal angle (Anterior Chamber IOL—ACIOL) to the fixation of the IOL to the iris or the sclera in the posterior chamber (Posterior Chamber IOL- PCIOL). These techniques are all burdened by high postoperative astigmatism and a slow visual recovery, due to the large incision needed for IOL insertion [10].

Recently, suture-free scleral fixation techniques are becoming the first choice for numerous cataract surgeons. These approaches combine the advantages of the conventional trans-scleral fixation procedures with some new benefits. Suture-free scleral fixation techniques may be suitable for almost all patients who need a secondary IOL because they do not require the integrity of the iris diaphragm; it must be also considered that the employment of foldable lenses require smaller incisions. Sutureless approaches also allow the possibility of avoiding complications related to sutures (suture-induced inflammation, suture erosions and knot exposure, IOL dislocations due to broken sutures), reducing bleeding events and endophthalmitis [8,9].

We retrospectively analyzed the visual and refractive outcomes of two different sutureless scleral fixation IOLs, Carlevale FIL-SSF IOL (Soleko, Italy) and three-piece IOL (Sensar AR40, Johnson & Johnson, New Brunswick, New Jersey, USA), for secondary IOL implantation in patients affected with aphakia or spontaneous IOL subluxation/luxation. Currently, the advancement of technologies and surgical procedures in cataract surgery contributes to the achievement of increasingly accurate refractive results, with higher patient expectations on visual recovery and less tolerance to refractive errors. In this scenario it would be desirable to have secondary IOL implantation procedures capable of ensuring the best possible refractive outcomes. In this scenario, the development of sutureless secondary IOL implant techniques ensures better refractive outcomes than IOL implant techniques that require wide corneal incisions and corneal and scleral sutures.

Carlevale lenses (Soleko, Italy) are innovative IOLs that assure anchoring to the scleral tissue by means of special harpoons without the need of sutures. Sensar AR40 (Johnson & Johnson, New Brunswick, New Jersey, USA) is a three-piece lens made for a sulcus or in-the-bag implant. We analyzed the refractive results of the sutureless secondary implant of these two IOLs and found not significant differences.

Correct positioning and alignment of the IOL are fundamental for visual quality and it is well known how IOL tilt can decrease optical performance of the eyes and induce defocusing, astigmatism, halo, glare and higher-order aberrations [11]. A previous meta-analysis reported that a 2–3° tilt is common after uneventful cataract surgeries and it is clinically unnoticed for any design of IOL (either spherical or aspherical): this could be considered the amount of IOL tilt tolerated [12].

Different methods for IOL tilt measuring are available. In this study, we used the one proposed by Yamane: IOL tilt angle was considered as the average of the IOL tilt angle in the horizontal and vertical images recorded with SA-OCT [2]. In our study, IOL tilt angle was, on average, 2.75° ± 1.47 in group 1 (Carlevale lens) and 2.51° ± 1.80 in group 2 (three-piece IOL), without statistically significant differences (*p* > 0.05). This is in agreement with previous studies: Barca et al. reported values of 2.08° ± 1.19 [13] whereas Fiore et al. values 2.2° ± 1.6 after SSF with the Carlevale lens [14]; Yamane registered a mean IOL tilt of 2.3° ± 1.9 with sutureless intrascleral fixation of a three-piece Tecnis IOL (ZA9003) [7].

We considered as significant a tilt angle mean value > 5°: this cut off is reported in literature as visually significant [11,15] after cataract surgery with in-bag implantation. However, we did not find a statistical correlation between the tilt degree and BCVA. A significant IOL tilt occurred in 7 eyes with the Carlevale lens and 6 eyes with three-piece lens, 25% and 24% respectively, without statistically significant difference. Correct alignment of the IOL is fundamental for avoiding IOL tilt and decreasing of visual performances. To avoid torsion, the haptics of the IOL shou ld be perfectly symmetrical: in both the surgical techniques, for the implant of either the Carlevale lens or 3-piece IOL implantation, we marked the limbus precisely at 0-180 degrees for the achievement of IOL centration. Moreover, the manipulation of the haptics is necessary for IOL centration, and the resulting deformation may contribute to unpleasant/unwanted IOL movements. The Carlevale lens has a specific design: the T-shape of the haptics and the soft material of the lens has been projected to reduce intraoperative manipulation and IOL decentration. On the other hand, in the 3-piece IOL implantation, some expedients may give the same results: marking the sclera 1.8 mm from the limbus for the sclerotomies execution; using vitreoretinal forceps for the externalization of the haptics, in order to prevent their elongation and deformation; the creation of bulbs at the end of the haptics, to provide greater stability inside the scleral tunnel; IOL haptic placement beneath the scleral flap, to prevent further movement of the haptics. Thus, these precautions may give the same results without increasing the surgical time.

IOL tilt has negative impact on visual performance by inducing optical aberrations. We evaluated the quality of the imaging system using the ocular PSF measured with Shack-Hartmann aberrometer (Osiris). PSF is the image of a point of light source projected onto the retina: the degree of blurring in the image is a measure for the quality of the imaging system. PSF is affected by aberrations: previous studies indicated that internal coma has been the most frequent type of aberration associated with IOL tilt [11,16]. In our study, mean values of PSF in the two groups didn’t show statistically significant differences. We achieved good refractive results in both treatment groups: spherical equivalent value was −0.33 D in group 1 and −0.35 D in group 2, without statistically significant differences. We found similar results in literature: D’Agostino et al. compared sutureless scleral fixation with the three-piece Alcon MA60AC and the Carlevale lens, finding no statistically significant differences in terms of Corrected Distance Visual Acuity (CDVA) and refractive errors [17].

Both the Carlevale lens and the 3-piece AR40 are a foldable lens and require small incisions for their implantation. Thus, both procedures induce low astigmatism, allowing rapid visual recovery. We observed a low cystoid macular edema rate in our series (two patients in both groups). We can speculate that the absence of iris contact, allows to avoid pigment dispersion and, consequently, chronic inflammation [18]. All cystoid macular edema cases were successfully treated with topical NSAIDs for 3 weeks.

Concerning the risk of corneal edema, we experienced bullous keratopathy in one case of a Carlevale lens implantation, that required endothelial keratoplasty, whereas two patients with 3-piece IOL had transient corneal edema which regressed with topical corticosteroid and hyperosmolar eyedrops. We attributed a certain endothelial loss to intraoperative manipulation, but in our study, we didn’t analyze pre- and post-operative endothelial cell count. Finally, only one patient with a Carlevale lens had haptic exposure after nine months, that required a new suture of the overlying conjunctiva: in this case the scleral flap was thinner than expected. None in the other group had this complication: thus, a correct creation of the scleral flap is fundamental and should be considered the procedure of choice. We are aware that our study presents some potential limitations. The small sample size and the retrospective design with follow-up data limited to one year probably represent the most important limitations. Nevertheless, the sample was large enough to allow valid statistical analysis between the two treatment groups. On the other hand, strengths of the study are the repeatability of the surgeries, since they were performed by one surgeon with standardized procedures, and the originality of the topic, because to our knowledge this is the first study which compares treatment visual and refractive results between the Carlevale lens and the three-piece AR40 IOLs for secondary IOL implantation. There is only one study previously published about sutureless scleral fixation techniques comparing the Carlevale lens and the three-piece MA60AC (Alcon); the authors concluded that the first IOL is better in terms of astigmatism (0.67 D ± 0.88 in group Carlevale vs. 1.91D ± 2.07 in group MA60AC, *p* = 0.04) and dislocation rate (0% in group Carlevale and 33% in group MA60AC, *p* = 0.01), whereas they did not find statistically significant differences in CDVA, mean refractive error and surgical time [17]; we believe that the results may be due to the intrinsic characteristics of the two 3-pieces IOLs, and this should be further analyzed. Moreover, the surgical technique of implanting differs because MA60AC IOLs were implanted with a different technique according to Yamane, thus the results are not comparable with ours. We believe that future studies may help to clarify these differences.

## 5. Conclusions

In conclusion, our results show that suture-free scleral fixation of either the Carlevale lens or the three-piece AR40 IOL for secondary IOL implantation have similar visual, refractive, and surgical outcomes, making them both viable options in this setting. Our study presents strengths and limits, as said above. Certainly, a prospective design with a bigger sample size could make the results more reliable; in addition, a longer follow-up may highlight late complications not evident after 12 months. This study provides some preliminary data that could be useful for further studies with more patients and longer follow-up.

## Figures and Tables

**Figure 1 jcm-12-00188-f001:**
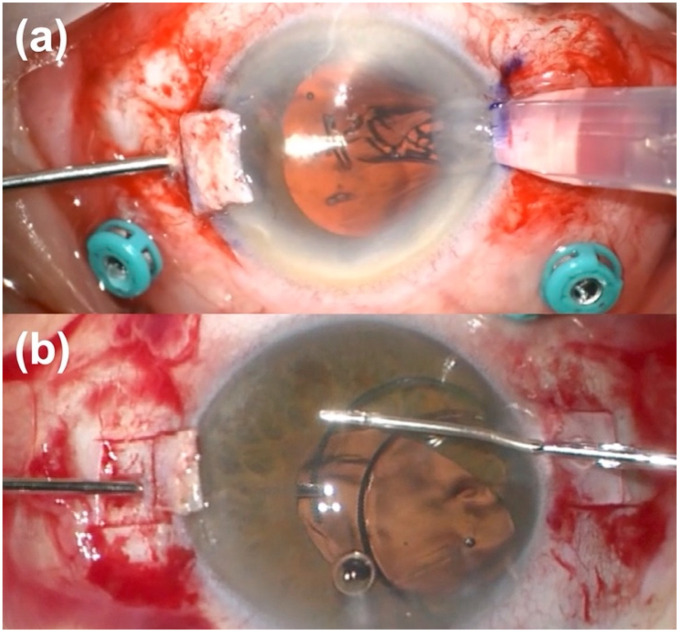
Intraoperative photos of suture-free scleral fixation intraocular lens implantation: (**a**) Carlevale lens, (**b**) suture-free three-piece lens. A carefully manipulation of the haptics is fundamental to avoid their damages.

**Figure 2 jcm-12-00188-f002:**
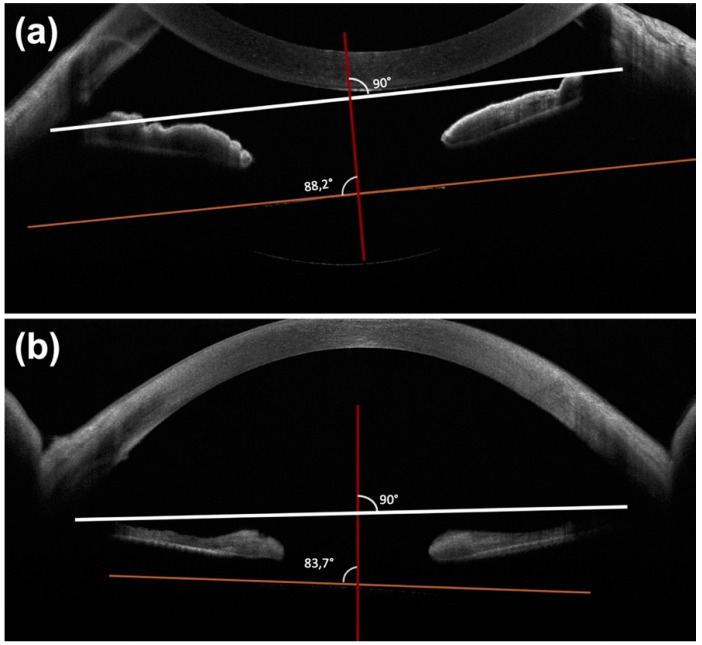
Anterior segment optical coherence scans. (**a**) The reference line was considered a straight line passing through the iridocorneal angle (white), and a second line was drawn on the horizontal axis of the IOL (orange). The angle between these two lines centered in correspondence to the corneal apex (red) was considered as the IOL tilt angle. The average of the IOL tilt angle in the horizontal and vertical images was recorded. (**b**) IOL tilt angle >5°.

**Table 1 jcm-12-00188-t001:** Demographic and clinical characteristics of patients included in the study.

	Carlevale IOL (Group 1)	Three-Piece IOL (Group 2)	*p*-Value
**Number of patients (eyes)**	28 (28)	25 (25)	
**Age (years), mean value ± SD**	75.2 ± 12.1	74.7 ± 16.7	0.81 *
Sex			
Male, *n* (%)	19 (67.9%)	13 (52%)	0.24 **
Female, *n* (%)	9 (32.1%)	12 (48%)	
**Clinical status**			
Aphakia, *n* (%)	3 (10.7%)	2 (7.14%)	0.92 **
Spontaneous luxation of IOL-capsular bag, *n* (%)	20 (71.4%)	19 (78.58%)	
IOL subluxation, *n* (%)	5 (17.9%)	4 (14.28%)	
**Preoperative spherical equivalent (D), mean value**	+9.32	+8.54	<0.001 *
**Baseline BCVA (logMAR), mean value ± SD**	0.78 ± 0.37	0.81 ± 0.31	0.59 *

BCVA = best corrected visual acuity. * Student’s t-test. ** Chi-square test.

**Table 2 jcm-12-00188-t002:** Postoperative results at different timepoints and comparison between the Carlevale group and the three-piece group.

	Carlevale IOL (Group 1)	Three-Piece IOL (Group 2)	*p*-Value ***
**Postoperative BCVA (logMAR), mean value**			
Month 1	0.32	0.49	0.37
Month 3	0.23	0.32	0.46
Month 6	0.23	0.32	0.45
Month 12	0.23	0.32	0.45
**IOL tilt angle, mean value ± SD**			
Month 1	2.86 ± 1.81	2.72 ± 1.65	0.73
Month 3	2.87 ± 1.47	2.70 ± 1.70	0.71
Month 6	2.75 ± 1.47	2.52 ± 1.90	0.68
Month 12	2.75 ± 1.47	2.51 ± 1.80	0.66
**Postoperative spherical equivalent (D), mean value**			
Month 1	−0.62	−0.77	0.86
Month 3	−0.43	−0.53	0.77
Month 6	−0.36	−0.32	0.75
Month 12	−0.33	−0.35	0.89
**Postoperative 3 mm PSF, mean value ± SD**			
Month 1	0.36 ± 0.09	0.31 ± 0.09	0.56
Month 3	0.22 ± 0.09	0.32 ± 0.12	0.66
Month 6	0.19 ± 0.09	0.18 ± 0.29	0.54
Month 12	0.18 ± 0.09	0.15 ± 0.05	0.75

BCVA = best corrected visual acuity; PSF = point spread function. * Student’s *t*-test.

**Table 3 jcm-12-00188-t003:** Postoperative complications.

	Carlevale IOL (Group 1)	Three-Piece IOL (Group 2)
**Transient corneal edema, *n* (%)**	1 (3.6%)	3 (12%)
**Bullous keratophaty, *n* (%)**	1 (3.6%)	0 (0%)
**Haptic exposure, *n* (%)**	1 (3.6%)	0 (0%)
**Macular edema, *n* (%)**	2 (7.1%)	2 (8%)

## Data Availability

The data presented in this study are available on request from the corresponding author. The data (original imaging) are not publicly available due to privacy issues.

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
