# Peer review of "Comparison of Visual and Aberrometric Outcomes in Suture-Free Scleral Fixation: Three-Piece Lenses versus Carlevale Lenses"

_jcm, 2022, doi:10.3390/jcm12010188_

Round 1
Reviewer 1 Report
Dear author,
Congratulations on the completion of this research work.
Here are some possible modifications to be made to the manuscript:
Introduction.
- Describe the characteristics of the two types of intraocular lenses used in the study (lines 73 and 79).
- Provide objective data on the prevalence/incidence of aphakia after cataract surgery and/or lens subluxation requiring secondary intervention without subsequent capsular support.
- Increase the number of references cited.
- Make clear the study hypothesis as well as the objective at the end of the introduction.
Materials and Methods.
- Please clarify how the sample size calculation was performed.
Results.
- Do not write repetitive information that is already in the tables. Provide only key information.
Discussion.
- Increase the number of references that allow you to compare your results with those in the literature.
- Discuss possible future projects or improvements to be made to the study.
References.
- Increase the number of bibliographical references.
Best regards,
Author Response
Reviewer 1 | 27 Nov 2022
Dear author,
Congratulations on the completion of this research work.
Here are some possible modifications to be made to the manuscript:
Introduction.
- Describe the characteristics of the two types of intraocular lenses used in the study (lines 73 and 79).
- Provide objective data on the prevalence/incidence of aphakia after cataract surgery and/or lens subluxation requiring secondary intervention without subsequent capsular support.
- Increase the number of references cited.
- Make clear the study hypothesis as well as the objective at the end of the introduction.
Materials and Methods.
- Please clarify how the sample size calculation was performed.
Results.
- Do not write repetitive information that is already in the tables. Provide only key information.
Discussion.
- Increase the number of references that allow you to compare your results with those in the literature.
- Discuss possible future projects or improvements to be made to the study.
References.
- Increase the number of bibliographical references.
Best regards,
Response: Thank you very much for the review of our manuscript. We sincerely appreciate all valuable comments and suggestions, which helped us to improve the quality of the article. Our responses are described below in a point-to-point manner.
Introduction.
- We modified the Introduction section by adding the requested information (description of the two types of lens included in the study, data on prevalence/incidence of aphakia after cataract surgery and/or lens subluxation requiring secondary intervention without subsequent capsular support, study hypothesis and objective).
Materials and Methods.
- This was a retrospective study on patients treated with suture-free scleral fixation IOLs (Carlevale and AR40). We included all patients treated with these procedures in the indicated time period. Exclusion criteria consisted of high refractive errors (myopia > 6 diopters, hypermetropia or astigmatism > 3 diopters), pupil diameter > 4.5 mm, significant ocular comorbidities, such as corneal lesions, retinal diseases, uveitis, and glaucoma.
Results
- We modified the Results section as suggested.
Conclusions
- We added a comment on possible future projects or improvements to be made to the study.
References
- We increased the number of cited references as suggested.

Reviewer 2 Report
Is it an interesting study, very curent, with a short follow-up, perriod but with significant results.
Review paper jcm-2080491
The objective of this study is interesting.
1. Introduction. Interesting interesting study, even if retrospective, comparing two ways of correcting aphakia
2. Materials and Methods. The criteria are clearly outlined and the determination methods are explained in detail. Postoperative clinical data, including postoperative complications occurrence, have been collected at the following time points after surgery: 1 month, 3 months, 6 months and 12 months. PSF was measured at post-operative visits at month 3 and 12. Very important is the use of oct anterior segment to evaluate the centering of the IOL
3. Results. . The results obtained are similar in the two groups. A significant IOL tilt (angle > 5°), occurred at a similar percentage within the two groups. There were no statistically significant differences (p-value>0.05) for all comparisons. A single 1case with haptic exposure in group 1 (Carlevale IOL). A small number of cases present postoperative complications. An important limitation is the fact that the results are only after one year, it is difficult to appreciate what happens with a series of complications that may appear: decentration, haptie exposure, secondary macular edema etc
4. Discussion. Due to the multitude of complicated cataracts that have appeared recently, the need to have an implant that offers good functional results, its stability and minor complications is increasingly important. Being recent implants, the results from the specialized literature do not give us a clear comparison of them. studies with longer follow-up periods are needed. do you have the possibility to show more recent studies?
5. The conclusions : the conclusions are based on the results obtained. Can be improved
Author Response
Reviewer 2 | 2 Dec 2022
Is it an interesting study, very curent, with a short follow-up, perriod but with significant results.
Review paper jcm-2080491
The objective of this study is interesting.
- Introduction. Interesting interesting study, even if retrospective, comparing two ways of correcting aphakia
- Materials and Methods. The criteria are clearly outlined and the determination methods are explained in detail. Postoperative clinical data, including postoperative complications occurrence, have been collected at the following time points after surgery: 1 month, 3 months, 6 months and 12 months. PSF was measured at post-operative visits at month 3 and 12. Very important is the use of oct anterior segment to evaluate the centering of the IOL
- Results. . The results obtained are similar in the two groups. A significant IOL tilt (angle > 5°), occurred at a similar percentage within the two groups. There were no statistically significant differences (p-value>0.05) for all comparisons. A single 1case with haptic exposure in group 1 (Carlevale IOL). A small number of cases present postoperative complications. An important limitation is the fact that the results are only after one year, it is difficult to appreciate what happens with a series of complications that may appear: decentration, haptie exposure, secondary macular edema etc
- Discussion. Due to the multitude of complicated cataracts that have appeared recently, the need to have an implant that offers good functional results, its stability and minor complications is increasingly important. Being recent implants, the results from the specialized literature do not give us a clear comparison of them. studies with longer follow-up periods are needed. do you have the possibility to show more recent studies?
- The conclusions : the conclusions are based on the results obtained. Can be improved
Response: Thank you for your helpful suggestions and positive comments.
We improved the Discussion and Conclusion section as you suggested.

Round 2
Reviewer 1 Report
Congratulations on the review. It improves on the previous version.
However, referencing other work done with similar sample sizes would lend credibility to the results.
Limitations and future projects should be reflected in the last paragraph of the discussion.
Best regards
Author Response
Reviewer 1 | 14 Dec 2022
Congratulations on the review. It improves on the previous version.
However, referencing other work done with similar sample sizes would lend credibility to the results.
Limitations and future projects should be reflected in the last paragraph of the discussion.
Best regards
Response: Thank you very much for the review of our manuscript. We added and discussed a previously published study as you suggested. We also discussed limitations and future projects in the discussion. Thank you for your helpful suggestions.